# Synthesis of Sodium Cobalt Fluoride/Reduced Graphene Oxide (NaCoF_3_/rGO) Nanocomposites and Investigation of Their Electrochemical Properties as Cathodes for Li-Ion Batteries

**DOI:** 10.3390/ma14030547

**Published:** 2021-01-24

**Authors:** Jiwoong Oh, Jooyoung Jang, Eunho Lim, Changshin Jo, Jinyoung Chun

**Affiliations:** 1School of Chemical Engineering and Materials Science, Chung-Ang University (CAU), 84 Heukseok-ro, Dongjakgu, Seoul 06974, Korea; shanall@cau.ac.kr (J.O.); wndud2362@cau.ac.kr (J.J.); 2Chemical & Process Technology Division, Korea Research Institute of Chemical Technology (KRICT), 141 Gajeongro, Daejeon 34114, Korea; 3Energy and Environmental Division, Korea Institute of Ceramic Engineering and Technology (KICET), Jinju, Gyeongnam 52851, Korea

**Keywords:** sodium cobalt fluoride, reduced graphene oxide, nanocomposites, solvothermal synthesis, Li-ion batteries, high-capacity cathodes

## Abstract

In this study, sodium cobalt fluoride (NaCoF_3_)/reduced graphene oxide (NCF/rGO) nanocomposites were fabricated through a simple one-pot solvothermal process and their electrochemical performance as cathodes for Li-ion batteries (LIBs) was investigated. The NCF nanoclusters (NCs) on the composites (300–500 nm in size) were formed by the assembly of primary nanoparticles (~20 nm), which were then incorporated on the surface of rGO. This morphology provided NCF NCs with a large surface area for efficient ion diffusion and also allowed for close contact with the conductive matrix to promote rapid electron transfer. As a cathode for LIBs, the NCF/rGO electrode achieved a high reversible capacity of 465 mAh·g^−1^ at 20 mA·g^−1^ via the conversion reaction, and this enhancement represented more than five times the reversible capacity of the bare NCF electrode. Additionally, the NCF/rGO electrode exhibited both better specific capacity and cyclability within the current density testing range (from 20 to 200 mA·g^−1^), compared with those of the bare NCF electrode.

## 1. Introduction

In the modern world, Li-ion batteries (LIBs) are essential energy storage devices. Due to the growing need for electric vehicles and energy storage systems, the demand for advanced LIBs has increased significantly in recent years. Accordingly, several studies on cathode materials [1,2,3], anode materials [4,5,6], separators [7,8], electrolytes [9,10], and other cell components [11] have been conducted, with a focus on manufacturing high-performance LIBs.

Intercalation materials such as LiCoO_2_ (LCO) and LiNi_x_Mn_y_Co_z_O_2_ (NMC) are actively used as cathode materials in commercial LIBs. They can promote highly reversible charge/discharge processes over stable cathode potential ranges. However, their Li^+^ storage capacity needs to be enhanced for the development of next-generation battery systems. Though LCO has demonstrated good cyclic stability in small-sized commercial LIBs, only about half of the theoretical capacity (274 mAh·g^−1^) is utilized owing to the structural strain [12]. Similarly, NMC has insufficient capacity for next-generation high-energy vehicle applications, even though it exhibits stable operating voltage and a good electrical conductivity (10^−5^ S·cm^−1^) [13]. To achieve high energy density, high capacity or wide reaction potential cathode materials are required. However, intercalation-based materials have intrinsically low reversible capacities owing to the limited space available for Li^+^ insertion into the materials. Therefore, to improve the energy density of LIBs, conversion reaction-based cells are considered.

Conversion materials are based on utilizing the multiple electrons that are present in the oxidation state of transition metals in metal-anion compounds (M_a_X_b_; M: transition metal such as Fe, Cu, and Co. X: anion such as O, F, S, and N), through a conversion reaction [14,15]. For cathode applications, studies on Fe_a_X_b_ compounds have been reported extensively [16,17,18,19], but Co_a_X_b_ compounds have been barely studied. Co_a_X_b_ compounds can be converted into hydride, sulfide, nitride, and fluoride forms [20]. Considering that cathode materials should have a high reaction voltage, the ionicity of the M-X bond must be increased to obtain higher output voltages [21]. Among the various Co_a_X_b_ compounds, only fluoride is suitable as a cathode material because its ionicity is significantly higher than that of the other compounds.

Sodium metal fluoride (NaMF_3_) which has large theoretical capacities and a large voltage window are promising cathode materials because they are thermodynamically stable under standard conditions, in contrast to LiMF_3_ [22]. Sodium cobalt fluoride, NaCoF_3_ (NCF), has a theoretical capacity of 578 mAh·g^−1^ in the reaction where 3 Li^+^ ions are involved (NaCoF_3_ ↔ Na^+^ + CoF_3_, CoF_3_ + 3Li^+^ + 3e^−^ ↔ Co + 3LiF) and 385 mAh·g^−1^ in the reaction where it reacts with 2 Li^+^ ions (NaCoF_3_ + 2Li^+^ + 2e^−^ ↔ NaF + Co + 2LiF). Although only two Li^+^ ions react in the latter case, the capacity is twice the capacity that is achieved with intercalation materials (i.e., LCO and NMC). However, several problems are associated with NCF, which include: (i) they have insulating properties due to their large bandgap. This low electrical conductivity can result in uneven distribution and slow diffusion of electrons in the electrode [23]; (ii) the metal formed by the conversion process should be reoxidized during the reverse process [24,25]. However, in the bulk state of some metal fluoride, significant separation of the metal and LiF occurs, resulting in a difficult reoxidization process [19]. For these reasons, only NaFeF_3_ has shown limited capacity (225 mAh·g^−1^) in sodium-ion battery (SIB) system within the voltage range for the intercalation reactions, and other NaMF_3_ materials (M = Co, Ni, Mn) have exhibited very low capacities (~33 mAh·g^−1^) [22].

Nanostructuring with conductive materials can improve cell performance because it promotes: (i) a large surface area for effective contact with the electrolyte and low areal current density; (ii) facile diffusion of electrons/ions in the nanosized composite structure; (iii) homogeneous dispersion of nanosized particles on the conductive materials, which helps to lower the resistance between NCF and the current collector. Therefore, we synthesized NCF nanoclusters (NCs) that comprised NCF nanoparticles (NPs) homogeneously dispersed on the conductive surface of reduced graphene oxide (rGO). The rGO helps to electron transport between the current collector and NPs [26,27,28]. Moreover, NPs are well dispersed on rGO, because rGO has a large surface area and functional groups interacting with NPs [29,30,31]. The NCF NPs assembled to form NCs on the rGO and prevented self-aggregation of the NCF and rGO nanosheets [32].

Studies have reported the use of ball milling and roll-quenching processes for the preparation of nanosized NaMF_3_ [22,33,34]. However, the NPs formed using these methods are not uniformly sized. When the size of particles is not uniform, the active materials of the electrode layer show non-uniform reaction kinetics, resulting in low capacity under high current densities [35]. When the capacity of ball-milling synthesized NaFeF_3_ was tested in a SIB, the electrode delivered a capacity of 169 mAh·g^−1^ in the potential range of 2–4.3 V [33]. However, it resulted in high charge/discharge polarization and poor cycle stability. In addition, NCF prepared by ball milling exhibits a rather low capacity of 38 mAh·g^−1^ in the potential range from 2 to 4.6 V in SIBs. Dimov et al. also fabricated nanosized NCFs using the ball milling and roll-quenching method. However, they obtained a capacity of 33 mAh·g^−1^ [22].

In this study, we synthesized NCF/rGO nanocomposites using a simple one-pot solvothermal process and investigated their electrochemical properties as cathodes for LIBs. A solvothermal process produces chemical compounds under high temperature and pressure using ethylene glycol as the solvent, leading to making uniform NCs assembled by NPs under optimized synthetic conditions. The process has the advantage of achieving precise control over the particle sizes and high-purity NCF (almost no presence of measured impurities). Furthermore, the particles obtained from this method have a large surface area for Li^+^ insertion because 20 nm-sized NPs are gathered to form uniform spherical clusters with a diameter of 300–500 nm. The electrical conductivity of the synthesized NPs can be improved by synthesizing rGO composite structures. As a cathode material for LIBs, the NCF/rGO electrode exhibited a high specific capacity (465 mAh·g^−1^ at 20 mA·g^−1^) through the conversion reaction. Additionally, the NCF/rGO electrode demonstrated a significantly higher capacity, which was five times the capacity of the electrode composed of bare NCF NCs.

## 2. Experimental

### 2.1. Synthesis of The NCF/rGO Nanocomposite

NCF/rGO nanocomposites were prepared by modifying a previously reported procedure [36]. First, GO (0.15 g) (Nanografi Co. Inc., Thuringia, Germany) was added in 40 mL of ethylene glycol (99.5%, Samchun Chemicals, Seoul, Korea) and dispersed using an ultrasonic liquid processor (VCX750, Sonics & Materials, Newtown, CT, USA). Ultrasonic dispersion was carried out for 30 min at 20 kHz with a nominal power of 750 W and 40% acoustic amplitude. In this process, jacketed beaker and cooling circulator were used to prevent overheating. Cobalt chloride hexahydrate (CoCl_2_·6H_2_O, 0.88 g) was dissolved separately in 20 mL of ethylene glycol using magnetic stirrer. Subsequently, the two solutions were mixed. Then, 0.2 g of trisodium citrate dihydrate (Na_3_Cit) and ammonium fluoride (NH_4_F) solution (0.411 g of NH_4_F dissolved in 2 mL of distilled water) were added sequentially to the mixture and it was stirred for 30 min. Next, 1.2 g of sodium acetate (NaOAc) was added under vigorous stirring. The resulting solution was transferred into a 100 mL Teflon-lined stainless-steel autoclave (JNT, Pohang, Korea). The autoclave was heated at 200 °C for 12 h and then naturally cooled to room temperature. After heating treatment, the products were separated by centrifugation at 5000 rpm for 5 min and washed with ethyl alcohol several times. The washed products were dried at 80 °C overnight. The synthesis method that was used to obtain the NCF NCs was similar to that used for the NCF/rGO nanocomposites, but GO-dispersed ethylene glycol was not used.

### 2.2. Materials Characterization

Powder X-ray diffraction (XRD) patterns were obtained using a Rigaku D/Max 2500/PC diffractometer (Rigaku Co., Tokyo, Japan) at a scanning rate of 6.00°·min^−1^. The morphologies of the materials were investigated using scanning electron microscopy (SEM; JSM-7000F, JEOL Ltd., Toyko, Japan, acceleration voltage: 10.0 kV) and transmission electron microscopy (TEM; JEM-2000EX, JEOL Ltd., Toyko, Japan, acceleration voltage: 200 kV). The Brunauer–Emmett–Teller (BET) surface area of the samples (Tristar II 3020, Micromeritics Inc., Norcross, GA, USA) was obtained from the nitrogen physisorption isotherms at 77 K. Pore size distributions were calculated from the adsorption branch of the isotherms using the Barrett–Joyner–Halenda (BJH) method. Raman spectroscopy was recorded on a Horiba Jobin-Yvon LabRam Aramis (Horiba, Kyoto, Japan) using laser excitation at 514 nm from an argon ion laser (<100 μW, ×50, and 300 s). Thermogravimetric analysis (TGA; Q600 SDT, TA instruments) was conducted at a heating rate of 10 °C·min^−1^ under airflow of 100 mL min^−1^. The carbon content of the samples was determined by elemental analysis using a CS 744 carbon/sulfur analyzer (LECO Corporation, St. Joseph, MI, USA). X-ray absorption near-edge structure (XANES) spectra were collected from the BL10C beam line at the Pohang light source (PLS-II, Pohang, Korea).

### 2.3. Electrochemical Measurements

For electrode preparation, the NCF/rGO nanocomposite was mixed with super P carbon and polyvinylidene difluoride (PVDF) at a weight ratio of 8:1:1 in N-methyl-2-pyrrolidone (NMP). The resulting slurry was coated onto an aluminum foil and dried at 100 °C in a vacuum. The dried electrodes were used to assemble coin cells (CR2032), using lithium foil as the counter and reference electrodes, and 1.0 M LiPF_6_ in ethylene carbonate/dimethyl carbonate (EC/DMC; 1:1 volume ratio, Enchem Ltd., Jaechun, Korea) as the electrolyte. The galvanostatic charge/discharge analyses were conducted using a battery cycler (WBCS3000L, WonATech Co., Seoul, Korea) within a current rate range of 20–200 mA·g^−1^. The specific capacity and current rate calculations were based on the mass of NCF. Considering the mass ratio between NCF and rGO was 85:15, NCF control sample electrode was mixed with super P carbon and PVDF at a 7:2:1 ratio in NMP; the amount of super P carbon was controlled to obtain similar carbon contents in each electrode.

## 3. Results and Discussion

### 3.1. Structural and Morphological Characterization

NCF/rGO nanocomposites were synthesized using the solvothermal process (Figure 1). CoCl_2_·6H_2_O (dissolved in ethylene glycol) and NH_4_F (dissolved in distilled water) were used as the cobalt and fluorine precursor solutions, respectively. Next, Na-containing precursors (NaOAc and Na_3_Cit, acting as the alkali source and electrostatic stabilizer) and graphene oxide were added to ethylene glycol. Na_3_Cit made NCF NCs uniformly shaped because of the strong coordination affinity of its multi-donating ligands with cobalt ions, which was consistent with previous reports [36,37]. In addition, an appropriate reaction temperature was crucial to obtain the NCF NCs of the target size without impurities during the solvothermal process [36]. The GO added to the solvothermal reaction was reduced to rGO using ethylene glycol as a solvent and reducing agent.

NCF NCs were evenly distributed and combined with rGO, which may be explained as follows: (i) GO exhibits a quasi-two-dimensional carbon nanosheet structure with a high surface area. Hence, it can be used as a support material for loading of NCF NCs; (ii) the NCs were grown on the surface of graphene sheets owing to the interactions between the functional groups on the GO surface, such as carboxyl (-COOH) and hydroxyl (-OH) [38,39].

Figure 2a shows the XRD pattern of the NCF/rGO nanocomposites prepared by the solvothermal method. This pattern corresponds to a pure orthorhombic phase of NCF, which is consistent with the JCPDS card No. 70–1889. The diffraction peaks 2θ of 22.7, 32.3, 38.4, 46.5, 52.4, and 58.5° were assigned to (101), (121), (031), (202), (222), and (123), respectively. The average crystal size of these NPs was 17.6 ± 0.5 nm, as calculated by the Debye–Scherrer equation. The diffraction pattern of GO showed peaks at 2θ of approximately 9.6 and 19.4° (Appendix A). For the NCF/rGO nanocomposites, these peaks disappeared and a broad peak located at approximately 20° was observed, indicating that GO was flaked and reduced. Impurity peaks corresponding to CoO_x_, NaF, and CoF_2_ were not present (Figure 2a and Appendix A).

The morphology and size of the NCF/rGO nanocomposites were observed by SEM and TEM. The SEM images in Figure 2b,c illustrate the morphology of the NCF/rGO nanocomposites, comprised of uniform spherical NCF NCs (300–500 nm in diameter) and sheets of rGO. While NCF NCs maintained their original NCs structure (Appendix A), the composites were evenly composed of rGO. As shown in the TEM images in Figure 2d, the NCF NCs, covered with rGO, were formed through the assembly of ~20 nm NPs, which is almost the same as the crystallite size calculated by XRD. The large surface area of the NCs facilitated their contact with the electrolyte. Furthermore, NPs can react rapidly with electrons at an applied current, which improves ion diffusion by decreasing the solid-state ion diffusion distance. In addition, nanosized pores that were formed both inside the NCs (between each NP) and between the NCF NCs and rGO, stabilized the active material during the charge/discharge processes, during which large volume changes occur typically.

Raman spectroscopy analysis confirmed the reduction of GO to rGO during the solvothermal reaction (Figure 3a). The two samples exhibited large G and D-bands. The I_D_/I_G_ intensity ratio increased from 0.92 (GO) to 1.05 (NCF/rGO). This difference explained the decrease in average size of the sp^2^ domain after reduction of the exfoliated GO and the removal of oxygen groups from the GO surface [40,41]. TGA and elemental analysis were performed to quantify the amount of rGO in the NCF/rGO nanocomposites (Appendix A). Excluding impurities, e.g., moisture, the weight ratio of NCF:rGO in the composite was estimated to be 85:15.

To investigate the surface area and pore volume of the NCF/rGO nanocomposite, we obtained nitrogen physisorption isotherms (Figure 3b). The surface area of the NCF/rGO nanocomposites was 34 m^2^·g^−1^ as calculated by the BET method, and the pore volume of the NCF/rGO nanocomposite was 0.146 cm^3^·g^−1^. Nitrogen adsorption increased significantly from a P/P_0_ ratio of 0.9, indicating that a few tens of nanosized pores were formed. Further, Figure 3b (inset) shows the BJH pore size distribution. This broad pore size distribution corresponds to pores between the NCF NPs, those between the rGO sheets and the NCF NPs, and those between the rGO sheets. Moreover, nanosized pores can both improve the electrolyte wettability through capillary condensation and stabilize the electrode upon volume changes due to stress.

### 3.2. Electrochemical Performance

Appendix A shows the galvanostatic charge/discharge profiles of the NCF/rGO nanocomposites obtained with 1 M LiPF_6_ in EC/DMC solution. The test was conducted to confirm whether desodiation (NaCoF_3_ ↔ Na^+^ + CoF_3_) was possible or not by increasing the voltage up to 5 V. The charge/discharge profile for NCF/rGO at 0.1 C, in the potential window of 2.5–5 V shows that a capacity of 188 mAh·g^−1^ is obtained when the voltage was increased to 5 V in the first cycle. This capacity could be obtained (although there was no Li^+^ in the cathode) owing to the following reasons: (i) a desodiation reaction may have occurred; (ii) anions (PF_6_^−^) would have influenced the capacity at the high-voltage range by adsorption on the rGO surface [42,43]; (iii) an irreversible electrolyte-decomposition reaction may have occurred. To confirm that desodiation hypothetically occurred, X-ray absorption spectroscopy was conducted to verify changes in the oxidation number of Co. XANES analysis in Appendix A shows that the peak position of pristine NCF/rGO nanocomposites is measured at approximately 7725 eV. This was consistent with the peak positions of Co^2+^ that have been reported previously [44,45]. In addition, there was negligible change in the red line (after 5 V charging) and the black line (before the electrochemical reaction). This indicated that the oxidation number was unchanged and hence desodiation did not occur. This may be due to the structural stability of NCF. A higher voltage is required to induce desodiation, but it is difficult to achieve this using a commercial electrolyte. Therefore, the first charging capacity of 188 mAh·g^−1^ was due to the irreversible electrolyte-decomposition reaction at high voltages. The first discharging capacity remarkably decreased to 25 mAh·g^−1^, which further supports the occurrence of the irreversible reaction process of NCF in the initial cycle.

Although desodiation did not occur, capacity due to a conversion reaction with Li^+^ can be obtained by decreasing the reaction operating voltage range. Figure 4 shows the charge/discharge profiles of the NCF/rGO nanocomposite electrode, measured in the potential range of 1–4.5 V at different current densities. In the first cycle, the specific capacity was 514 mAh·g^−1^ at a discharge voltage (lithiation) of 1 V with a current density of 20 mA·g^−1^. A voltage plateau of less than 2 V was observed, which was similar to the conversion reaction. Therefore, the conversion reaction of NCF occurred (2Li^+^ + NaCoF_3_ ↔ Co + 2LiF + NaF) [46]. In the first cycle, a long voltage plateau was observed at 1.21 V. However, in the second discharge cycle, the voltage plateau reduced in length as the voltage increased. This indicated that the initial lithiation process led to the rearrangement of the atomic structures via the conversion reaction. Moreover, structural changes following the first conversion occurred during the first cycle. The first charge capacity of the NCF/rGO electrode was 465 mAh·g^−1^ (initial coulombic efficiency of 90.5%). The reversible capacity of 465 mAh·g^−1^ is the highest among the reported data for NCF in literature. At a current density of 20 mA·g^−1^, the capacity exceeded the theoretical capacity of NCF (385 mAh·g^−1^). This can be explained as follows: (i) additional Li^+^ might have been stored on the surface of rGO or anions (PF_6_^−^) might have been adsorbed on the rGO surface [43]; (ii) owing to the nanosized particles, the corresponding surface area increased, resulting in interfacial Li^+^ insertion into the NCF NCs [47,48].

In the first cycle, the specific discharge capacity was measured to be 514, 363, 297, and 119 mAh·g^−1^ at the current densities of 20, 50, 100, and 200 mA·g^−1^, respectively. NCF/rGO exhibited an initial discharge capacity of 297 mAh·g^−1^ even at a current of 100 mA·g^−1^. This will be compared with the NCF electrode results.

As NCF has poor electrical conductivity, the NCF electrode (without the rGO supporting structure) could not achieve high capacities (Figure 5). In the first cycle, when the discharge voltage was 1 V, the specific capacities were 186, 139, 92, and 11 mAh·g^−1^, respectively, at different current densities (from 20 to 200 mA·g^−1^). The reversible capacity of the NCF electrode was ~89 mAh·g^−1^ at 20 mA·g^−1^. As we confirmed in electrochemical analysis, although both electrodes have similar carbon content in the electrode layer, NCF/rGO electrode exhibited higher capacity (465 mAh·g^−1^ at 20 mA·g^−1^), less polarization, and improved reversibility compared to those of NCF electrode. It can be inferred that rGO with highly dispersed 3-D structures played a crucial role in increasing the reversible capacities of the NCF electrodes owing to its high conductivity and enhanced contact with NCF NPs [49].

Figure 6 shows the cycling performance of the NCF/rGO and NCF electrodes at 100 mA·g^−1^. The NCF/rGO electrode exhibited higher capacities than the NCF electrode (without rGO) in all the cycles. The 50th cycle capacities of the NCF/rGO and NCF electrodes were 111 and 24 mAh·g^−1^, respectively. Although capacity reduction was observed for the NCF/rGO electrode, higher capacities were obtained, compared with the NCF electrode. Figure 7 shows the cycling performance of the cobalt fluoride-based materials at various current densities [22,50,51,52,53,54,55]. Interestingly, this is the first report showing promising cyclability of NCF. Moreover, NCF/rGO exhibits similar cyclability compared to some CoF_2_ cathodes. The capacity decrease observed for the NCF/rGO electrode can be attributed to Co dissolution at high voltages and stability decay due to changes in the particle structures during the conversion reaction [46,56]. In particular, as voltage increased during delithiation process, a portion of metallic Co is oxidized and easily dissolved as Co ions into the electrolyte [46]. Therefore, although the NCF/rGO electrode exhibited higher capacities than the practical capacities of commercial cathode materials, such as LCO (~140 mAh·g^−1^) and NMC (160–200 mAh·g^−1^), at the initial stages, it is essential to improve the cyclability of NCF/rGO (cf. commercial cathode materials exhibit more than 80% capacity retention over hundreds of cycles). For future research, we can propose that the stability and thus the cell performance can be enhanced by introducing a fine coating layer on each NCF particle surface [57,58].

## 4. Conclusions

In summary, we synthesized NCF/rGO nanocomposites through a simple one-pot solvothermal process. The obtained NCF/rGO nanocomposites comprised 300–500 nm sized NCF NCs, which were assembled from NPs (~20 nm) incorporated on the surface of rGO (exhibiting excellent electrical conductivity). The nanosized particles resulted in a more efficient electrochemical reaction and enhanced ion diffusion. Moreover, through the uniform arrangement of the rGO sheets, NCF/rGO nanocomposites with high surface area (34 m^2^·g^−1^) were obtained. As a cathode material for LIBs, the NCF/rGO electrode exhibited a high reversible capacity of 465 mAh·g^−1^ at 20 mA·g^−1^ in the conversion reaction, which is five times higher than the reversible capacity of the NCF electrode. Moreover, the NCF/rGO electrode exhibited enhanced specific capacity within the entire current density range (20–200 mA·g^−1^), compared with the performance of the NCF electrode. This study can contribute to metal fluoride research, demonstrating that this material can be a promising cathode for high energy density batteries.

## Figures and Tables

**Figure 1 materials-14-00547-f001:**
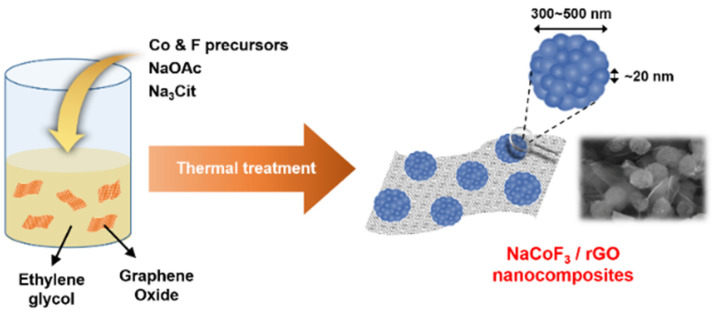
Schematic of the solvothermal synthesis of NCF/rGO nanocomposites.

**Figure 2 materials-14-00547-f002:**
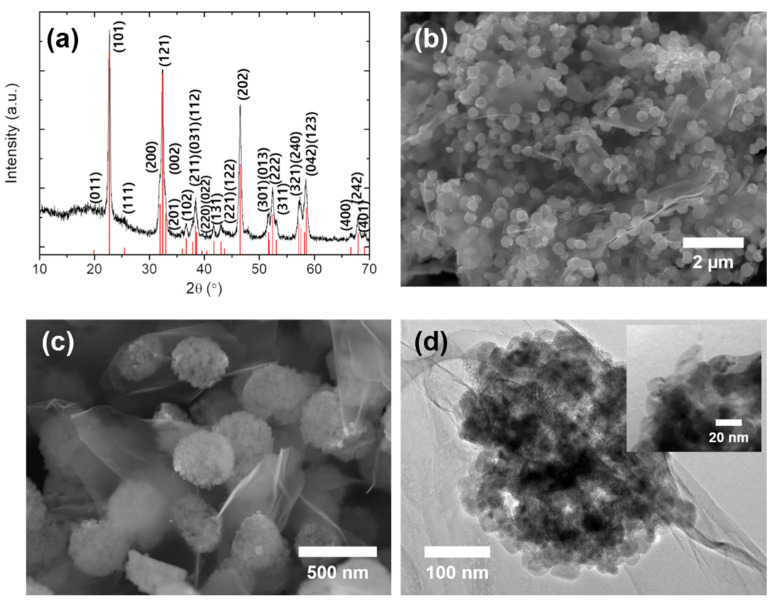
(**a**) XRD pattern (red columns: the standard peaks of NaCoF_3_, JCPDS No. 70-1889), (**b**,**c**) SEM images, and (**d**) TEM images of the NCF/rGO nanocomposites (inset: high-magnification the TEM image).

**Figure 3 materials-14-00547-f003:**
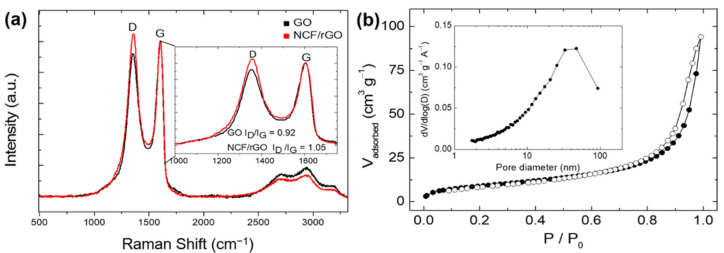
(**a**) Raman spectra of the GO and NCF/rGO samples; (**b**) Nitrogen physisorption isotherm and Barrett–Joyner–Halenda (BJH) pore size distribution (inset) of the NCF/rGO nanocomposite.

**Figure 4 materials-14-00547-f004:**
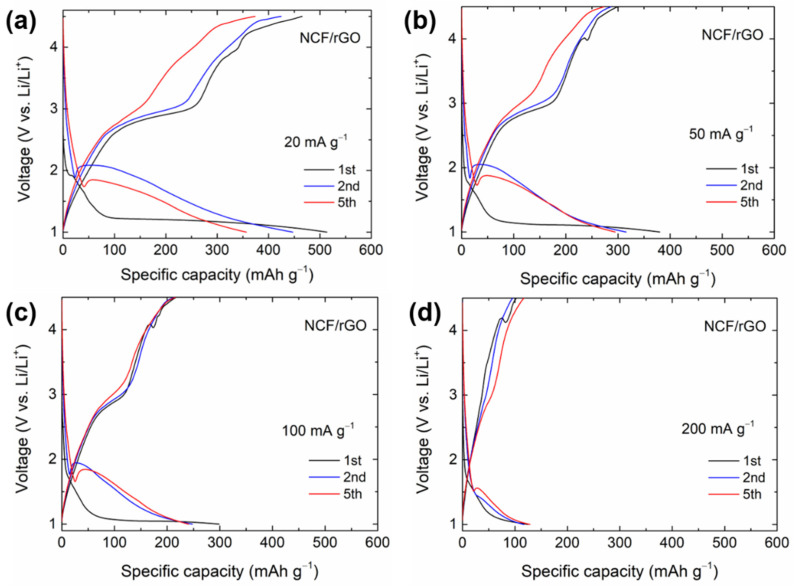
Galvanostatic charge/discharge curves of the 1st, 2nd, and 5th cycles for the NCF/rGO electrode at a current density of (**a**) 20 mA·g^−1^, (**b**) 50 mA·g^−1^, (**c**) 100 mA·g^−1^, and (**d**) 200 mA·g^−1^_._

**Figure 5 materials-14-00547-f005:**
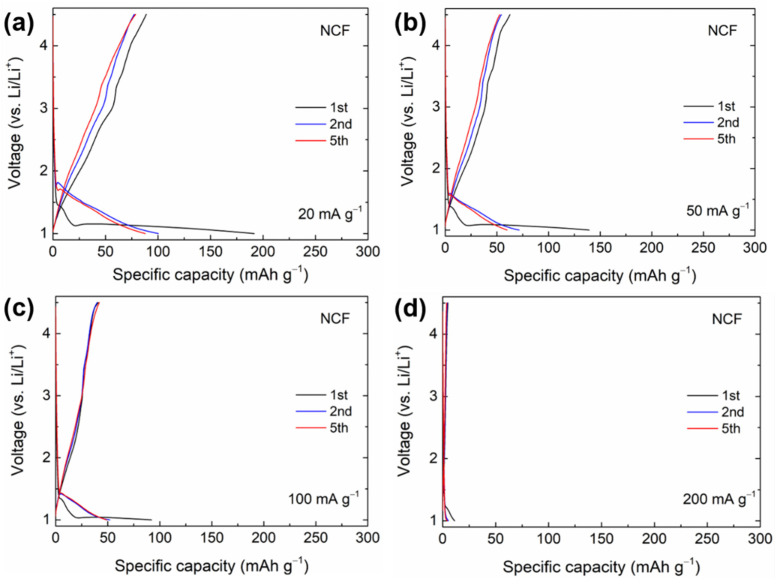
Galvanostatic charge/discharge curves of the 1st, 2nd, and 5th cycles for the NCF electrode at a current density of (**a**) 20 mA·g^−1^ (**b**) 50 mA·g^−1^ (**c**) 100 mA·g^−1^, and (**d**) 200 mA·g^−1^.

**Figure 6 materials-14-00547-f006:**
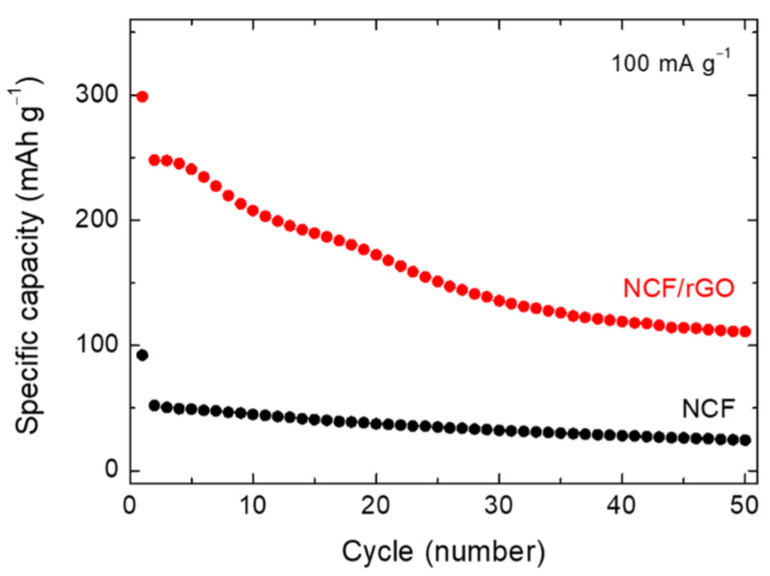
Cycling performance for the NCF/rGO and NCF electrodes at a current density of 100 mA·g^−1^.

**Figure 7 materials-14-00547-f007:**
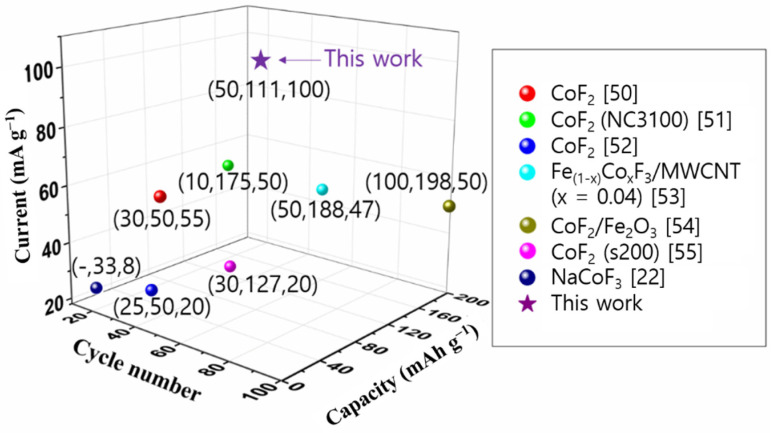
Cycle number vs capacity vs current of cobalt fluoride-based materials in LIB. The represented points are based on previous studies (x,y,z x: cycle number y: capacity, z: current).

## Data Availability

The data presented in this study are available on request from corresponding authors.

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
