# Peer review of "Synthesis of Sodium Cobalt Fluoride/Reduced Graphene Oxide (NaCoF3/rGO) Nanocomposites and Investigation of Their Electrochemical Properties as Cathodes for Li-Ion Batteries"

_materials, 2021, doi:10.3390/ma14030547_

Round 1

Reviewer 1 Report

In this study by Oh et al., the authors combined sodium cobalt fluoride with reduced graphene oxide and investigated their electrochemical performance. The article is quite interesting and definitely fits the scope of Materials. However, there are certain issues highlighted below, which should be handled before the paper can be recommended for publication.
1) Sonication due to high power cavitation may impair the properties of the material, therefore it is always important to specify the parameters of such processing. These are missing in the evaluated manuscript, so please supplement at least device name, power, and amplitude settings in the revised version of the manuscript.
2) What was the acceleration voltage for SEM and TEM imaging? This can influence the depth of penetration by the electron beam, thereby affecting the quality of the micrographs.
3) Specific information regarding the employed parameters for Raman spectroscopy characterization should also be added.
4) Airflow in the TGA description is missing.
5) Overall, many more details are absent, which makes it impossible to replicate this article. Consequently, other scientists cannot verify these results and build on them, which limits the impact of this contribution. Please screen the whole file for missing details and include them to solve this problem.
6) Article formatting should be improved:
- Fig. 2a is too small to read
- the appearance of plots is not consistent - e.g. plots in Fig. 3 are much larger
- inset to Fig. 3a is not visible and to Fig. 3b is barely visible
7) The benefit of using rGO is not clear from the evaluated data. Please conduct an additional experiment in which instead of graphene oxide, graphite would be used as a filler. It is essential to validate if nanostructured carbon gives some sort of synergy with NaCoF3 or it does not really matter what is put inside as long as the conductivity is improved by incorporation of more conductive material and the surface chemistry of the two components enables good integration of the composite.

Reviewer 2 Report

The manuscript named as "Synthesis of sodium cobalt fluoride/reduced graphene oxide (NaCoF3/rGO) nanocomposites and investigation of their electrochemical properties as cathodes for Li-ion batteries" is an innovative research paper that utilizes a nanocomposite structure as cathode and the results are promising. I suggest that it can be accepted. One minor suggestion is that the standard XRD pattern can be added into Figure 2a. 

Reviewer 3 Report

The present manuscript deals the “Synthesis of NaCoF3/rGO nanocomposites and their investigation as cathodes for Li-ion batteries”. The authors synthesised the nanocomposites and characterized by XRD, SEM, TEM, BET, Raman spectroscopy, XANES and finally performed the electrochemical measurements. They also compared the electrochemical properties of nanocomposite with pure NCF. This is a well written manuscript with meaningful results and in line with the scope of the journal. The authors did excellent job in explaining the effect of nanocomposite of NCF with rGO.  I recommend for a minor revision.

My comments are appended below.

1) The weight ratio of NCF:rGO was estimated to 85:15 using TGA. How this ratio was determined?

2) The cycling performance of NCF/rGO is higher than that of NCF, however, the capacity decrease is observed. The authors should compare the capacity of NCF/rGO with other well-known cathodes and a discussion should be included in the manuscript.

Round 2

Reviewer 1 Report

Thank you for following the suggestions. I am more than happy to recommend the publication of this paper in Materials